# On the Consistency of Maximum Likelihood Estimation of Probabilistic Principal Component Analysis

**Arghya Datta**[*]
Université de Montréal
Department of Mathematics and Statistics

**Sayak Chakrabarty**[†]
Northwestern University
Department of Computer Science

## Abstract

Probabilistic principal component analysis (PPCA) is currently one of the most used statistical tools to reduce the ambient dimension of the data. From multidimensional scaling to the imputation of missing data, PPCA has a broad spectrum of applications ranging from science and engineering to quantitative finance.

Despite this wide applicability in various fields, hardly any theoretical guarantees exist to justify the soundness of the maximum likelihood (ML) solution for this model. In fact, it is well known that the maximum likelihood estimation (MLE) can only recover the true model parameters up to a rotation. The main obstruction is posed by the inherent identifiability nature of the PPCA model resulting from the rotational symmetry of the parameterization. To resolve this ambiguity, we propose a novel approach using quotient topological spaces and in particular, we show that the maximum likelihood solution is consistent in an appropriate quotient Euclidean space. Furthermore, our consistency results encompass a more general class of estimators beyond the MLE. Strong consistency of the ML estimate and consequently strong covariance estimation of the PPCA model have also been established under a compactness assumption.

## 1 Introduction

In the era of big data, principal component analysis (PCA) is a standard dimension reduction tool frequently used in exploratory data analysis. PCA is however not a statistical model, implying that uncertainty quantification is not available. Uncertainty quantification is desirable in some situations, like when principal components have an interpretation. Probabilistic principal component analysis (PPCA) is a model introduced by Tipping and Bishop [1999] to remedy the situation. It consists of an additive model with a normal noise term. Let $p > q$ be two given positive integers. The PPCA model can be written as $x = \mathbf{W}z + \varepsilon$, where $x \in \mathbb{R}^p$ denotes a random vector that is observed after a linear transformation $\mathbf{W}$ is applied to a latent (hidden) variable $z \in \mathbb{R}^q$ up to a gaussian noise $\varepsilon \sim \mathcal{N}(0, \sigma^2 I_p)$. We assume that our data is centered, meaning $x$ has zero mean. In practice, the latent variable $z$ is not observed and is conventionally assumed to be normally distributed. Given a set of $n$ data points or observations $\{x_1, x_2, \ldots, x_n\} \subset \mathbb{R}^p$, the two unknown parameters $\mathbf{W}$ and $\sigma^2$ are estimated by maximum likelihood estimation (MLE).

PPCA is a very well-known and pervasive technique used in dimension reduction of high-dimensional data and used in figuring out the relevant dimensions in large-scale data mining. It must be said that

---

[*]arghya.datta@umontreal.ca

[†]sayakchakrabarty2025@u.northwestern.edu

37th Conference on Neural Information Processing Systems (NeurIPS 2023).

the primary goal of PPCA is not always to perform better than traditional PCA but open up a series of possibilities for future extensions for further analysis. PPCA also enables comparison with other probabilistic techniques. Namely, it is possible to quantify the uncertainty in parameters $\mathbf{W}$ and $\sigma^2$ through a Bayesian reformulation [Tipping and Bishop, 1999]. One can also consider modelling the error terms using heavy-tailed distributions (such as Student-t) to achieve robustness [Gai et al., 2008]. These models have appealing properties; in particular, the quality of the information retained from the original data is similar to that of the original PCA with added flexibility.

Despite the wide applicability, nothing is known about the consistency of the maximum likelihood (ML) estimation of the PPCA. The main obstruction is posed by the identifiability of the PPCA model over the parameter space $(\mathbf{W}, \sigma^2) \in \mathbb{M}(p, q) \times \mathbb{R}_+$, where $\mathbb{M}(p, q)$ and $\mathbb{R}_+$ denote the set of all $p \times q$ matrices and the set of positive real numbers respectively. One can restrict the attention only to rank $q$ matrices, but as we shall see, doing so will not significantly make our goal any harder (or easier) to achieve as the ML estimation of $\mathbf{W}$ has rank $q$. It can be readily seen that our model induces a Gaussian distribution on the data points $x \sim \mathcal{N}(0, \mathbf{W}\mathbf{W}^T + \sigma^2 I_p)$, which is to be used for the maximum likelihood estimation. From a frequentist point of view, we now assume that the unknown parameters $(\mathbf{W}, \sigma^2)$ has a true value which is equal to $(\mathbf{W}_0, \sigma_0^2)$. Additionally, we observe that the marginal distribution of the data points $\mathcal{N}(0, \mathbf{W}\mathbf{W}^T + \sigma^2 I_p)$ used for the ML estimation remains invariant under any translation of $\mathbf{W}$ by an orthogonal matrix $\mathbf{R}$ since $(\mathbf{W}\mathbf{R})(\mathbf{W}\mathbf{R})^T = \mathbf{W}\mathbf{W}^T$. As can be seen in the literature it is only possible to recover the maximum likelihood estimate of $\mathbf{W}$ up to rotations [Tipping and Bishop, 1999]. This lack of identifiability puts an immediate roadblock to directly applying Wald's consistency theorems [Wald, 1949], a well-known method to guarantee the consistency of maximum likelihood estimate.

It turns out that the quotients of the Euclidean spaces and in particular, the quotient of the space $\mathbb{M}(p, q) \times \mathbb{R}_+ \subset \mathbb{R}^{p \times q} \times \mathbb{R}$ is the natural topological object to consider to talk about the consistency results of the PPCA model. In a simplified situation when $\sigma^2$ is known, heuristically, the quotient space treats all the points $\mathbf{W}_0$ and its rotational translates $\mathbf{W}_0\mathbf{R}$ as a single entity by labelling them equivalent and therefore throwing all of them into a single equivalence class. Therefore, the parameter space for $\mathbf{W}$, i.e. $\Sigma = \mathbb{M}(p, q)$ becomes a set of equivalence classes $\tilde{\Sigma} = \{[\mathbf{W}] : \mathbf{W} \in \mathbb{M}(p, q)\}$ through an equivalence relation $\mathbf{W}' \in [\mathbf{W}]$ iff $\mathbf{W}'\mathbf{W}'^T = \mathbf{W}\mathbf{W}^T$ or equivalently $\mathbf{W}' = \mathbf{W}\mathbf{R}$, for some orthogonal matrix $\mathbf{R}$. In this setting, we thus get rid of the identifiability issue by passing through the quotient of $\mathbb{R}^{p \times q} \times \mathbb{R}$ via the proposed equivalence relation. We then study the consistency of the maximum likelihood in this space. The formal details about the definition of quotient space by a given equivalence relation can be found in Section 4.

The above approach is inspired by Redner [1981] where the author outlines a way to extend Wald's consistency theorems [Wald, 1949] under the non-identifiability assumption. We would like to mention that, even though passing through the quotient topological space eliminates the rotational ambiguity of the ML estimate by definition, it introduces many further technical issues. To the best of our knowledge, all the subtleties have not been properly addressed in Redner's work even though the author applies the proposed methodology to analyze the consistency of ML estimates of mixture models. One of the key things that is not explicitly mentioned in their work is that the underlying metric space structure of the quotient topological space can be substantially different from the natural topological structure of the quotient space itself and the topology generated by the open balls with the induced metric in quotient space is in general, not same as that of the usual quotient topology. Thus related geometric properties, for instance, the notion of convergence and other regularity properties of the parameter space become more delicate to analyze. We address all these underlying issues carefully later in section 4.

To the best of our knowledge, no previous research work has attempted to justify the wide usage of the ML estimate of the PPCA model because of the inherent identifiability nature of the model parameters. To resolve this challenge we propose a novel framework to see the PPCA model through the lens of an appropriate quotient of an Euclidean space. This perspective not only allows us to formalize the problem in precise mathematical terms but also helps us resolve the dispute caused by the non-identifiability.

*Considering from the point of novelty, the contributions of this paper are two-fold:* we prove that the maximum likelihood estimates $(\hat{\mathbf{W}}, \hat{\sigma}^2)$ are consistent, i.e., $(\hat{\mathbf{W}}, \hat{\sigma}^2) \to (\mathbf{W}_0, \sigma_0{}^2)$ in probability

in the appropriate quotient Euclidean space. In fact, our consistency results are valid for a general family of estimators that in particular, contains the maximum likelihood estimator. Furthermore, strong consistency, i.e., an almost sure convergence of the ML estimates to the true parameter is also deduced when the parameter space is compact.

*The rest of the paper is organized as follows:* section 2 outlines a brief description of the necessary background followed by a comprehensive discussion of relevant research works. In section 3, we formally introduce the PPCA model, and the ML estimates of the parameters of interest and talk about the main question we are interested in. In section 4, we provide several basic definitions and mention a few key results from the theory of quotient topological spaces in a self-contained fashion. Section 5 contains all our contributions followed by their detailed proofs in section 6. In section 7, we state Wald's conditions and interpret them within the quotient parameter space. Finally, in section 8, we talk about the impact of our work and put forward a few open questions before concluding in section 9.

## 2 Background and Previous Work

The abstract formalism concerning quotient topological spaces is not foreign to the theory of applied statistics. For an in-depth discussion about the application of quotient space into various statistical models, see McCullagh [1999]. Even though, the necessary tools and definition of topological quotients are introduced in section 4, an inquisitive reader is certainly encouraged to consult the classic Munkres. The main line of attack used in this paper has been previously introduced in Redner [1981], where the author proves the consistency of mixture models under the standard regularity conditions of Wald's consistency theorems [Wald, 1949]. It is worthwhile to mention that the theoretical justification of using the maximum likelihood estimates in practice has gained a lot of recent interest. For instance, Chen [2017] has a rigorous treatment of several consistency results with an application to non-parametric mixture models.

In the previous section, we have already mentioned that PPCA offers many advantages over traditional PCA, particularly in terms of model flexibility and the potential for generalization to accommodate various statistical reformulations. Building upon the recognition of its strength by Bengio et al. [2013], Goodfellow et al. [2016], and Ruff et al. [2021] as one of the most notable advancements in probabilistic models, subsequent research endeavours have leveraged PPCA to achieve remarkable results in various applications. Within the realm of cryo-electron microscopy (cryo-EM) in the field of biology, extensive discussions in the literature have explored its efficacy, as evidenced by the works of Heimowitz et al. [2018], Penczek et al. [2011], Tagare et al. [2015]. Similarly, in the domain of computer vision, the probabilistic model has proven its superiority, as demonstrated by Szeliski [2022] and others. While the precise reasons underlying its success in certain situations remain elusive, PPCA has found widespread applicability in diverse domains. It has found great applications in incremental learning for visual tracking [Lim et al., 2004], in the study of Gaussian processes [Bonilla et al., 2007, Lawrence and Hyvärinen, 2005], orthogonal signal correction (OSC) [Lee et al., 2023], weighted nuclear norm minimization [Gu et al., 2017], and in outlier detection algorithms like [Domingues et al., 2018]. Recent advancements have showcased its potential in areas such as few-shot learning [Wang et al., 2020], data mining techniques [Witten and Frank, 2002], and most notably, in finance for exploiting market integration for pure alpha investments [Tzagkarakis et al., 2013]. The broad spectrum of applications where PPCA has exhibited remarkable performance underscores its versatility and its value as a powerful tool in numerous domains.

As we have already noted, the theoretical justification behind the ML estimates used for the PPCA model in practice is not always straightforward and has recently attracted the attention of several authors. In a fairly recent work [Chérief-Abdellatif, 2019], the authors provide a consistent estimator of the quantity $\mathbf{W}\mathbf{W}^T$ assuming $\sigma^2$ is known. Their work is primarily concerned with the model selection questions in statistics using a variational approach. Even though in our work we assume the rank of the matrix rank($\mathbf{W}$) $= q$ to be known and is fixed by the user (this assumption can be removed as long as $q$ does not grow with the sample size $n$), it is of independent interest to want to estimate $q$. In their work [Chérief-Abdellatif, 2019], the authors constructed an estimator of the

quantity $q$ to demonstrate how closely their estimated model recovers the true model corresponding to the true value of $\mathbf{W}$, they were also able to estimate the marginal covariance of the data points and thereby came up with a consistent estimator of $\mathbf{W}\mathbf{W}^T$. One may notice that, in this case, estimating $\mathbf{W}\mathbf{W}^T$ instead of $\mathbf{W}$ eliminates the non-identifiability issue. It is important to mention that their work does not use the maximum likelihood approach and instead uses variational techniques to construct an estimator of $\mathbf{W}\mathbf{W}^T$. In this regard, it is relevant to mention that another work [Bouveyron et al., 2011] was able to provide a consistent estimator of the parameter $q$ using maximum likelihood estimation. Finally, we restate that no related previous work attempts to justify the commonly used ML estimates $(\widehat{\mathbf{W}}, \widehat{\sigma}^2)$ of the PPCA model, perhaps due to ambiguity caused by the inherent non-identifiable parameter space. In the following section, we begin by presenting the PPCA model and concerned ML estimates originally developed in Tipping and Bishop [1999]

## 3 Problem Statement

**Notation** In what follows, we shall always let capital letters denote random variables and their deterministic realizations will be denoted by lowercase letters. Let $X_1, X_2, \ldots$ be a sequence of independent identically distributed random variables where the p.d.f. of the distribution of $X_1$ is known except a parameter $\theta = (\mathbf{W}_\theta, \sigma_\theta^2)$ in some parameter space $\Theta \subset \mathbb{R}^{p \times q} \times \mathbb{R}_+$ for some matrix $\mathbf{W}_\theta$ and positive integers $p, q$. We assume the distribution of $X_1$ is generated from a true parameter $\theta_0$. For a given parameter $\theta = (\mathbf{W}_\theta, \sigma_\theta^2)$, we let $f(x; \theta)$ denote the normal p.d.f. $\mathcal{N}(x; 0, \mathbf{W}_\theta \mathbf{W}_\theta^T + \sigma_\theta^2 I_p)$. Henceforth, the notation $\|\mathbf{W}\|$ will mainly refer to the spectral norm, which is defined as the largest eigenvalue of the matrix $\mathbf{W}\mathbf{W}^T$. It is worth noting that in some rare instances, we have employed the Frobenius norm, but such usage has been explicitly mentioned. Unless otherwise stated, all the probabilities and expectations are taken with respect to the true parameter $\theta_0$. Finally, for a $m \times m$ matrix $\mathbf{A}$, we denote its characteristic polynomial $\det(\lambda I_m - A)$ by $\mathcal{P}_A(\lambda)$.

**The PPCA model:** Let us consider that we have access to a structured dataset taking the form of a matrix $\mathbf{X}$ whose lines are given by $x_1, \ldots, x_n \in \mathbb{R}^p$ with $p$ a large positive integer. We assume that the columns of $\mathbf{X}$ are centered (meaning a mean of 0). We assume

$$x_i = \mathbf{W} z_i + \varepsilon_i, \tag{3.1}$$

where $z_1, \ldots, z_n, \varepsilon_1, \ldots, \varepsilon_n$ are i.i.d. the realizations of independent random variables with $z \sim \mathcal{N}(0, I_q)$ and $\varepsilon \sim \mathcal{N}(0, \sigma^2 I_p)$, and $\mathbf{W}$ is an unknown fixed $p \times q$ matrix, $\sigma > 0$ being an unknown fixed scale parameter and $I_q$ and $I_p$ being identity matrices of size $q$ and $p$, respectively. The random variable $z$ plays the role of a hidden (latent) variable and $\varepsilon$ that of an error term. We recall that rank$(\mathbf{W}) = q$. To estimate the parameters $\mathbf{W}$ (often called the loading matrix) and the unknown variance $\sigma^2$ of the homoscedastic error term $\epsilon$, Tipping and Bishop [1999] employs the maximum likelihood (ML) procedure and solutions are given as follows

$$\widehat{\mathbf{W}} = \mathbf{U} \left( \Delta_q - \sigma^2 I_q \right)^{1/2} \mathbf{R} \text{ and } \widehat{\sigma}^2 = \frac{1}{p-q} \sum_{j=q+1}^{p} \delta_j,$$

where $\mathbf{R}$ is any $q \times q$ rotation matrix, the columns of $\mathbf{U}$ are given by the first $q$ eigen vectors of the sample covariance matrix $S_x = 1/n \sum_{i=1}^{n} x_i x_i^T$ and $\Delta_q = \text{diag}(\delta_1, \ldots, \delta_q)$ are the $q$ eigenvalues of $S_x$ sorted in the descending order. We note that the solution $\widehat{\mathbf{W}}$, in this case, is not unique since $\mathbf{R}$ could be any orthogonal matrix which in practice, is often set to be $I_q$. For now, we redistrict our discussion assuming the rank $q$ is fixed. For this model, given the parameters $(\mathbf{W}, \sigma^2)$, we have the following marginal density for the generated data $\{x_i\}$

$$p(x|\mathbf{W}, \sigma) = \int p(x|z, \mathbf{W}, \sigma) p(z) dz = \mathcal{N}(x; 0, \mathbf{W}\mathbf{W}^T + \sigma^2 I_p),$$

where we used the fact that the latent variable $z$ is distributed as $\sim \mathcal{N}(0, I_q)$. Thus for our problem, the parameter space is

$$\Theta := \left\{ \theta : \theta = (\mathbf{W}_\theta, \sigma_\theta^2) \in \mathbb{R}^{p \times q} \times \mathbb{R}_+ \right\}.$$

In the above definition there is a slight abuse of notation as the parameter $\theta$ is self-referencing to itself. Nonetheless, we retain the notation for the sake of clarity and the ease of exposition. We assume the data points $\{x_i\}$ are generated from a true parameter $\theta_0 = (\mathbf{W}_0, \sigma_0^2)$, i.e., they are i.i.d. samples from normal distribution whose p.d.f. is given by $f(x; \theta_0)$. In this work, we aim to prove the consistency of maximum likelihood estimates, i.e. we want to discuss whether $(\widehat{\mathbf{W}}, \widehat{\sigma}^2) \to (\mathbf{W}_0, \sigma_0^2)$ holds $P_{\theta_0}$ almost surely or in probability.

Nearly all consistency proofs of well-known statistical models take identifiability for granted. Unfortunately, the PPCA model is not identifiable, since there may be matrices $\mathbf{W}_\theta$ and $\mathbf{W}_\phi$ such that $\mathbf{W}_\theta \mathbf{W}_\theta^T = \mathbf{W}_\phi \mathbf{W}_\phi^T$, in which case $f(x; \theta) = f(x; \phi)$ if $\sigma_\theta = \sigma_\phi$. It is not difficult to see in such cases, $\mathbf{W}_\phi$ will be a rotational translate of the matrix $\mathbf{W}_\theta$, i.e., $\mathbf{W}_\phi = \mathbf{W}_\theta \mathbf{R}'$ for some orthogonal matrix $\mathbf{R}'$. This rotational symmetry in parameterization results in a lack of identifiability. In the current work we adopt a similar approach as outlined in Redner [1981] and subsequently work in a quotient topological space of $\Theta$. Next, we rigorously introduce this quotient space framework and develop analytical tools to prove our consistency results.

## 4 A primer on quotient topological spaces

In this section, we give the definition of a quotient of a topological space by a given equivalence relation and discuss several important results pertinent to our work.

**Definition 4.1** (Equivalence relation)**.** Let $S$ be a set. A binary relation $\sim$ on $S$ is said to be an equivalence iff it is (i) reflexive: $x \sim x, \forall x \in S$, (ii) symmetric: $x \sim y \iff y \sim x, \forall x, y \in S$ and (iii) transitive: $x \sim y, y \sim z \implies x \sim z, \forall x, y, z \in S$.
It is evident that an equivalence relation on a set induces a partition of the set and vice-versa. The disjoint sets in this partition are called the equivalence classes. If $s \in S$ is given, then we write $[s] := \{y \in S \mid y \sim s\}$ for the equivalence class that contains $s$ and finally the set of equivalent classes $(S/\sim) := \{[s] \mid s \in S\}$ are called the **quotient** of the set $S$ by the equivalence relation $\sim$.

### 4.1 Quotient topological spaces and its metrizability

Given a space $X$ and an equivalence relation $\sim$ on $X$, the set-theoretic quotient $X/\sim$ (the set of equivalence classes) inherits a topology from $X$, called the quotient topology [Munkres]. It is well known that the surjective map $\pi : X \to X/\sim$ defined as $x \to [x]$ is continuous. We now record a useful result from Munkres, which will be important in proving our main results in the forthcoming sections.

**Theorem 4.2.** *If $X, X/\sim$ are topological spaces stated as above, then for another topological space $Y$ and for a continuous map $f : X \to Y$ with the additional property that $x \sim x'$ implies $f(x) = f(x')$, there exists unique continuous map $\tilde{f} : (X/\sim) \to Y$ such that $f = \tilde{f} \circ \pi$*

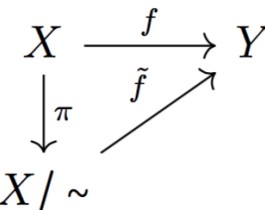

Figure 1: Illustration of the above theorem using a commutative diagram.

Throughout this paper, we have $Y = \mathbb{R}$ and $X$ is a subset of $\mathbb{R}^p$ for some positive integer $p$, which is a complete metric space and Euclidean distance serves as a natural metric on $\mathbb{R}^p$. The key idea is to extend this metric from $X$ to $X/\sim$ and this job is accomplished through an intermediate state, by introducing a suitable pseudometric on $X$. A pseudometric $\delta$ is a function on $X \times X$ to non-negative real numbers and the only difference between a pseudometric and a metric is that $\delta(x, y) = 0$ need not imply $x = y$ for $x, y \in X$. To construct the most commonly used metric on the quotient space $X/\sim$ we first introduce:

let $X \subset \mathbb{R}^p$ be a complete subspace of the Euclidean space and $d$ be the standard Euclidean distance on $\mathbb{R}^p$. For $x, y \in X$, we let

$$d_1(x, y) := \inf\{d(x_1, y_1) + \ldots + d(x_n, y_n) \mid x \sim x_1, y_i \sim x_{i+1}(1 \leqslant i \leqslant n-1), y_n \sim y\}.$$

Then it can be checked that $d_1$ defines a pseudometric on $X$. We now declare $x \approx y$ whenever $d_1(x, y) = 0$, and it is not difficult to verify the relation $\approx$ is an equivalence. Then we can endow the set-theoretic quotient $X/\approx$ with the metric $\bar{d}$, which we define as $\bar{d}([x], [y]) = d_1(x, y)$ where $[x], [y]$ are the classes of $x, y$ in $X/\approx$. Furthermore, we let $X_\approx$ to be the completion of the above space under the metric $\bar{d}$, meaning all the Cauchy sequences converge.

As the reader may have noticed already, there are two equivalence relations at play on $X$, namely $\sim$ and $\approx$. Although the construction of the space $X/\approx$ is dependant on the equivalence $\sim$ through the definition of $d_1$, it is not straightforward to identify the relationship between the two abstract quotient spaces $X/\sim$ and $X/\approx$. Now is a good time to pause and unveil the primary reason to justify the development of these technical tools. *Our initial goal is to metrize the topological quotient $X/\sim$ for a given equivalence relation $\sim$ so that we could extend Wald's conditions [Wald, 1949] in $X/\sim$ to prove consistency results in this space.* The procedure outlined in the previous paragraph provides a recipe for constructing a metric $\bar{d}$ on the space $X/\approx$. The next crucial result, Lemma 4.3 shows that for certain nice equivalence relations $\sim$, the two spaces $X/\approx$ and $X/\sim$ are identical. This result is a key fact that allows us to maneuver the metric construction into $X/\sim$. We now state the result from Weaver [2018]. The proof of this result is skipped as it is deemed to be technical and it serves a little purpose to our main goal. For a thorough treatment of metrizatiblity of quotient topological spaces, we refer the inquisitive reader to the comprehensive book on Lipschitz algebras [Weaver, 2018].

**Lemma 4.3.** *If $C$ is a closed subset of a complete metric space $X$. Let $X/C$ denote the quotient space of $X$ by the equivalence defined by $x \sim y$ if either $x = y$ or $x, y \in C$ then the underlying set $X_\approx$ and the quotient $X/C$ are the same. The metric on $X/C$ in this case, will take the form:*

$$\bar{d}([x], [y]) = \min(d(x, y), d(x, C) + d(y, C)) \quad \text{for any } [x], [y] \in X/C.$$

In the above statement, $d(x, C)$ denotes the distance between the point $x$ and the closed subset $C$ defined by $d(x, C) := \inf_{c \in C} d(x, c)$ and likewise for $d(y, C)$. An immediate follow-up question of the above result would be: what is the space $X$ and the closed subset $C$ we use in our context? We now recall two things from the previous section, first the parameter space for the PPCA model which is $\Theta \subset \mathbb{R}^{pq+1}$ and second the issue of identifiability, which arose because there could exist two parameters $\theta, \phi \in \Theta$ such that their corresponding densities are equal, which is equivalent to $\mathbf{W}_\theta \mathbf{W}_\theta^T + \sigma_\theta^2 I_p = \mathbf{W}_\phi \mathbf{W}_\phi^T + \sigma_\phi^2 I_p$. In what follows, using Redner's idea [Redner, 1981] we consider the closed subset defined by

$$C := \{\theta \in \Theta : \mathbf{W}_\theta \mathbf{W}_\theta^T + \sigma_\theta^2 I_p = \mathbf{W}_0 \mathbf{W}_0^T + \sigma_0^2 I_p\} \subset \Theta,$$

which allows us to get rid of identifiability since in the quotient topological space $\Theta/C$, $f(x; [\theta]) = f(x; [\theta_0])$ if and only if $[\theta] = [\theta_0]$ or equivalently $\theta \sim \theta_0$ by construction. We can now hope for checking Wald's consistency conditions in $\Theta/C$ to prove $[\hat{\theta}]$ converges to $[\theta_0]$ in probability, where $\hat{\theta} = (\hat{W}, \hat{\sigma}^2)$ denotes the MLE.

In the introduction, we pointed out that despite the extension of Wald's work [Wald, 1949] to quotient topological spaces by Redner [1981], an inadequate amount of attention was paid to the comprehensive understanding of the interplay between the quotient topology and the topology generated by the metric $\bar{d}$ in that treatment. For instance, a priori it is, by no means obvious how one would interpret their assumption 3 in the quotient space $\Theta/C$ without invoking Lemma 4.3. Also, their assumption 5 does not clearly mention what mode of convergence has been used since there are two available notions of convergence, one coming from the quotient topology and another coming from the metric topology generated by $\bar{d}$, and those two topologies are not same in general. It is a nontrivial fact that the latter is contained in the former (for a discussion on this topic see James [1990]). However, in most statistical applications, for example in the applications to mixture distributions and clustering [Redner, 1981], we tend to quotient the parameter space by a suitable closed subspace depending on the context, and subsequently the quotient metric takes an easier form as given in Lemma 4.3. Therefore, several technical constructions get greatly simplified in these applications.

# 5 Theoretical Results

In this section, we state our main results. Let $\{x_i\}_{i=1}^n \subset \mathbb{R}^p$ be a sequence of data points that are generated from latent observations $\{z_i\}_{i=1}^n \in \mathbb{R}^q$ via the PPCA model defined in (3.1) and recall from section 3 that the data points are assumed to be generated from the true parameter $\theta_0 := (\mathbf{W}_0, \sigma_0^2) \in \Theta$ of the underlying model. Our data points $x_i$ can therefore be viewed as a sequence of realizations of i.i.d. random variables whose true marginal density is given by $f(x; \theta_0) = \mathcal{N}(x; 0, \mathbf{W}_0\mathbf{W}_0^T + \sigma_0^2 I_p)$. We note a crucial point that for two parameters $\theta, \phi \in \Theta$, $f(x; \theta) = f(x; \phi)$ holds if and only if $[\theta] = [\phi]$ in $\Theta/C$. Therefore, we write $f(x; \theta)$ for simplicity, instead of cumbersome $f(x; [\theta])$ when stating results in the quotient space $\Theta/C$. In the spirit of Wald [1949] and Redner [1981], two of our main results are the following:

**Theorem 5.1.** *Let $S$ is any closed subset of $\Theta$ not intersecting $C$ then*

$$\mathbb{P}\left(\limsup_n \sup_{\theta \in S} \prod_{i=1}^n \frac{f(x_i; \theta)}{f(x_i; \theta_0)} = 0\right) = 1.$$

**Theorem 5.2.** *Let $\theta_n(x_1, \dots, x_n)$ be any measurable function of the observations $x_1, \dots, x_n$ such that*

$$\prod_{i=1}^n \frac{f(x_i; \theta_n)}{f(x_i; \theta_0)} \geqslant c > 0 \text{ for all } n,$$

*then $\mathbb{P}\left(\lim_n [\theta_n] = [\theta_0]\right) = 1$, that is $\theta_n$ converges to $\theta_0$ in $\theta/C$ almost surely, and therefore in probability.*

The above results have elegant geometric interpretations. In particular, Theorem 5.1 implies that if $U_C$ is any open neighbourhood containing $C$ then almost surely all but finitely many terms of the sequence $\{\theta_n\}$ land inside $U_C$. As an immediate consequence of Theorem 5.2 when $c = 1$, we obtain:

**Corollary 5.3** (Strong consistency of the MLE in the quotient space). *Let $\theta_n$ denote the sequence of maximum likelihood estimates of the PPCA model. We then have that*

$$\mathbb{P}\left(\lim_n [\theta_n] = [\theta_0]\right) = 1$$

*that is maximum likelihood estimates converge to the true parameter $\theta_0$ almost surely, and therefore in probability in the quotient space $\Theta/C$.*

Additionally, using the above corollary we obtain that the covariance for the PPCA process can be consistently (strong) estimated under a compactness assumption.

**Theorem 5.4.** *Let $\Theta_0$ be a compact subset of $\Theta$ containing the point $\theta_0 = (\mathbf{W}_0, \sigma_0^2)$. Let $\theta_n = (\widehat{\mathbf{W}}, \widehat{\sigma}^2)$ denote the sequence of maximum likelihood estimates of the PPCA model. Then we have*

$$\mathbb{P}\left(\widehat{\mathbf{W}}\widehat{\mathbf{W}}^T + \widehat{\sigma}^2 I_p \to \mathbf{W}_0\mathbf{W}_0^T + \sigma_0^2 I_p\right) = 1.$$

Now we talk about possible generalizations above results in light of Wolfowitz [1949], where the observations $\{X_i\}_{i=1}^n$ are assumed to be identically distributed but *need not be independent*. As noted in Wolfwowitz's work that Wald [1949] proves strong consistency (i.e. almost sure convergence) and his proof techniques can be extended for dependent random variables $\{X_i\}_{i=1}^n$ as long as the sequence $\{X_i\}$ satisfies the strong law of large numbers. In the spirit of Wald's work, Wolfowitz [1949] proves the consistency of ML estimates (i.e. convergence in probability) only under the assumption that the sequence $\{X_i\}$ satisfies weak law of large numbers. The assumption that the sequence $\{X_i\}$ satisfies the weak law of large numbers may seem technical but it offers a great deal of flexibility in real life applications due to the following result of Bernstein [Cacoullos, 2012].

**Theorem 5.5** (Bernstein). *Let $\{X_i\}$ be a sequence of centered random variables. If there exists a constant $\kappa > 0$ such that for every $i \in \mathbb{N}$ we have $Var(X_i) \leqslant \kappa$ and if the following condition is true:*

$$\lim_{|i-j| \to \infty} Cov(X_i, X_j) = 0,$$

*then the weak law of large numbers holds.*

We may note that the above result could be extremely relevant in real life applications as intuitively it accommodates all statistical models which allows local dependence among observations, in the sense that $X_i$ can significantly correlate with $X_j$ as long as $j$ remains sufficiently close to $i$ (i.e., $|i - j|$ is small), but the correlation gradually vanishes as $|i - j|$ becomes large. Temporal stochastic processes or time series data with appropriate assumptions could be tailored into the category of those statistical models where Bernstein's result applies. We therefore state our next set of results on the consistency of maximum likelihood estimates of the PPCA model but only under a weak law assumption. The following result is an immediate consequence of Wolfowitz [1949].

**Theorem 5.6.** *Let the data points $\{x_i\}_{i=1}^n$ be realizations of identically distributed random variables $\{X_i\}_{i=1}^n$ whose density is given by $f(x; \theta_0)$. Let the sequence of random variables $\{X_i\}$ satisfy the weak law of large numbers. Given $\eta > 0$ and a closed subset $S$ of $\Theta$ not intersecting $C$, there exists a quantity $h(S) \in (0, 1)$ which depends only on the closed set $S$ and an integer $N(\eta, S)$ such that*

$$\mathbb{P}\left(\sup_{\theta \in S} \prod_{i=1}^n \frac{f(x_i; \theta)}{f(x_i; \theta_0)} > h^n(S)\right) < \eta \text{ for every } n \geqslant N(\eta, S).$$

**Theorem 5.7.** *Let the data points $\{x_i\}_{i=1}^n$ be realizations of identically distributed random variables $\{X_i\}_{i=1}^n$ whose density is given by $f(x; \theta_0)$. Let the sequence of random variables $\{X_i\}$ satisfy the weak law of large numbers. Let $\theta_n(x_1, \ldots, x_n)$ be any measurable function of the observations $x_1, \ldots, x_n$ such that*

$$\prod_{i=1}^n \frac{f(x_i; \theta_n)}{f(x_i; \theta_0)} \geqslant c > 0 \text{ for all } n,$$

*then $[\theta_n] \xrightarrow{\mathbb{P}} [\theta_0]$, that is $\theta_n$ converges to $\theta_0$ in $\theta/C$ in probability.*

Furthermore we also have the two following results in spirit of corollary 5.3 and Theorem 5.4.

**Corollary 5.8** (Consistency of the MLE in the quotient space). *Let the data points $\{x_i\}_{i=1}^n$ be realizations of identically distributed random variables $\{X_i\}_{i=1}^n$ whose density is given by $f(x; \theta_0)$. Let the sequence of random variables $\{X_i\}$ satisfy the weak law of large numbers. Let $\theta_n$ denote the sequence of maximum likelihood estimates of the PPCA model. We then have that*

$$[\theta_n] \xrightarrow{\mathbb{P}} [\theta_0],$$

*that is maximum likelihood estimates converge to the true parameter $\theta_0$ in probability in the quotient space $\Theta/C$.*

Additionally, using the above corollary we can obtain prove consistent covariance estimation is possible for the PPCA model under a compactness assumption.

**Theorem 5.9** (Consistent estimation of covariance). *Let the data points $\{x_i\}_{i=1}^n$ be realizations of identically distributed random variables $\{X_i\}_{i=1}^n$ whose density is given by $f(x; \theta_0)$. Let the sequence of random variables $\{X_i\}$ satisfy the weak law of large numbers. Let $\Theta_0$ be a compact subset of $\Theta$ containing the point $\theta_0 = (\boldsymbol{W}_0, \sigma_0^2)$. Let $\theta_n = (\widehat{\boldsymbol{W}}, \widehat{\sigma}^2)$ denote the sequence of maximum likelihood estimates of the PPCA model. Then we have*

$$\widehat{\boldsymbol{W}}\widehat{\boldsymbol{W}}^T + \widehat{\sigma}^2 I_p \xrightarrow{\mathbb{P}} \boldsymbol{W}_0\boldsymbol{W}_0^T + \sigma_0^2 I_p.$$

## 6 Proof of theoretical results

We only present the proofs of Theorem 5.7 and Theorem 5.9. Corollary 5.8 follows from Theorem 5.7. The proofs of Theorem 5.2, corollary 5.3 and Theorem 5.4 will be analogous.

*Proof of Theorem 5.7.* We fix two quantities $\eta_1, \eta_2 > 0$. We need to show that there exists $N_0(\eta_1, \eta_2)$ such that

$$\mathbb{P}\left(\bar{d}([\theta_n], [\theta_0]) > \eta_1\right) \leqslant \eta_2$$

holds for all $n \geqslant N_0(\eta_1, \eta_2)$. We consider the following closed subset of the parameter space $\Theta$ which does not intersect $C$

$$S_0 := \left\{\theta \in \Theta : \bar{d}([\theta], [\theta_0]) \geqslant \eta_1\right\}.$$

Using Theorem 5.6 we select $0 < h(\eta_2) < 1$ such that

$$\mathbb{P}\left(\sup_{\theta \in S_0} \prod_{i=1}^{n} \frac{f(x_i; \theta)}{f(x_i; \theta_0)} > h^n(\eta_2)\right) < \eta_2 \text{ for every } n \geqslant N(\eta_1, \eta_2),$$

for some $N(\eta_1, \eta_2)$. The dependence on $\eta_1$ comes from the choice of the set $S_0$. Next, we choose $N_0$ such that $h^n(\eta_2) < c$ holds for every $n \geqslant N_0$. Let $N_0(\eta_1, \eta_2) = \max(N_0, N(\eta_1, \eta_2))$. Therefore for all $n \geqslant N_0(\eta_1, \eta_2)$, we have that

$$\mathbb{P}(\bar{d}([\theta_n], [\theta_0]) > \eta_1) \leqslant \mathbb{P}\left(\sup_{\theta \in S_0} \prod_{i=1}^{n} \frac{f(x_i; \theta)}{f(x_i; \theta_0)} \geqslant c\right)$$

$$\leqslant \mathbb{P}\left(\sup_{\theta \in S_0} \prod_{i=1}^{n} \frac{f(x_i; \theta)}{f(x_i; \theta_0)} > h^n(\eta_2)\right)$$

$$\leqslant \eta_2.$$

$\square$

To prove Theorem 5.9 we need the following lemma.

**Lemma 6.1.** *Let $Y$ be a complete metric space, and let $\psi : \Theta \to Y$ be a Lipschitz map such that $\psi(\theta) = \psi(\theta')$ whenever $\theta \sim \theta'$. Then $\psi$ lifts to an unique Lipschitz map from $\Theta/C$ to $Y$.*

*Proof.* The proof of the statement follows from proposition 1.4.4 and proposition 1.4.3 in Weaver [2018]. $\square$

*Proof of Theorem 5.9.* We consider the function $\psi : \Theta \to \mathbb{R}^{p \times p}$ defined by $\psi(\mathbf{W}, \sigma^2) = \mathbf{W}\mathbf{W}^T + \sigma^2 I_p$. Since $\theta_0 \in \text{int}(\Theta_0)$, we can therefore consider the restriction $\psi|_{\Theta_0} : \Theta_0 \to \mathbb{R}^{p \times p}$, which is a Lipschitz function as $\psi$ is $C^1$ and $\Theta_0$ is compact. Therefore the lift $\tilde{\psi}\Big|_{\Theta_0} : \Theta_0/C \to \mathbb{R}^{p \times p}$ is Lipschitz and hence continuous with respect to the topology generated by the metric $\bar{d}$.

Invoking corollary 5.8, we have $[\widehat{\mathbf{W}}, \widehat{\sigma}^2] \xrightarrow{\mathbb{P}} [\mathbf{W}_0, \sigma_0^2]$. Since $\tilde{\psi}\Big|_{\Theta_0}$ is continuous with respect to the topology generated by the quotient metric $\bar{d}$ from the previous lemma, using standard results from probability theory we infer that $\tilde{\psi}\Big|_{\Theta_0}([\widehat{\mathbf{W}}, \widehat{\sigma}^2]) \xrightarrow{\mathbb{P}} \tilde{\psi}\Big|_{\Theta_0}([\mathbf{W}_0, \sigma_0^2])$. $\square$

As we saw in the demonstration above, Lemma 6.1 plays a crucial role. We would like to *emphasize* that Theorem 4.2 would not be sufficient to ensure the continuity of the lift of $\psi$ with respect to the metric $\bar{d}$ as the quotient topology needs to be same that of the topology generated by the metric $\bar{d}$ on the quotient space $\Theta/C$ (see the counterexamples subsection in the appendices).

## 7  Auxiliary Lemmas

Our goal for this section will be to state Wald's conditions [Wald, 1949] to establish the consistency of MLE of the PPCA model and interpret those in the quotient parameter space $\Theta/C$ as stated in Redner [1981]. Unless otherwise stated, throughout this section we will be working in the quotient parameter space $\Theta/C$ introduced in section 4. We recall that $\theta_0 = (\mathbf{W}_0, \sigma_0^2)$ denotes the unknown parameter such that the true marginal distribution of the data points $\{x_i\}_{i=1}^{n}$ is given by the density $f(x; \theta_0) = \mathcal{N}(0, \mathbf{W}_0\mathbf{W}_0^T + \sigma_0^2 I_p)$. For a given $r > 0$, we let $N_r([\theta])$ denote the closed ball of radius $r$ around $[\theta]$ in $\Theta/C$. Following Redner [1981] we begin by introducing the quantities:

$$f(x, \theta, r) = \sup_{[\phi] \in N_r([\theta])} f(x; \phi) \text{ and } f^*(x, \theta, r) = \max\{1, f(x, \theta, r)\},$$

$$h(x, s) = \sup_{[\phi] \notin N_s([\theta_0])} f(x; \phi) \text{ and } h^*(x, s) = \max\{1, h(x, s)\}.$$

We are now in a position to state Wald's conditions as a series of following lemmas whose proofs will imply Theorem 5.1 and Theorem 5.6 as a consequence of the results stated in Wald [1949] and Wolfowitz [1949], respectively.

**Lemma 7.1.** *The parameter space $(\Theta/C, \bar{d})$ is a metric space with the property that every closed and bounded subset of $\Theta/C$ is compact.*

**Lemma 7.2.** *For each parameter $[\theta] \in \Theta/C$ and for sufficiently small $r$ and sufficiently large $s$, $f(., \theta, r)$ is measurable and the following expectations are bounded*

$$\mathbb{E}_{\theta_0}[\log f^*(x, \theta, r)] < \infty \text{ and } \mathbb{E}_{\theta_0}[\log h^*(x, s)] < \infty.$$

**Lemma 7.3.** *Let $\{[\theta_i]\} \subset \Theta/C$ be a sequence. If $\bar{d}([\theta_i], [\theta_0]) \to \infty$ then $f(x; \theta_i) \to 0$ except on a $P_{\theta_0}$-null set which does not depend on the sequence $\{[\theta_i]\}$.*

**Lemma 7.4.** *For each $([\theta], [\phi]) \in \Theta/C \times \Theta/C$ We have*

$$\mathbb{E}_{\phi}[|\log f(x; \theta)|] < \infty.$$

**Lemma 7.5.** *If $[\theta_i] \to [\theta]$ in $\Theta/C$ then $f(x; \theta_i) \to f(x; \theta)$ except on a $P_{\theta_0}$-null set which does not depend on the sequence $\{[\theta_i]\}$.*

The above conditions are fairly technical and primarily deal with the geometry of the quotient parameter space $\Theta/C$. One also needs to be careful as there are two different topologies available in this quotient space which need not be the same. For instance, in Lemma 7.5, the convergence under the if condition $[\theta_i] \to [\theta]$ is required to hold in the topology generated by the metric $\bar{d}$ which is a stronger requirement than merely asking for convergence in the quotient topology. The proofs of the above results are deferred to the relevant subsection in the appendices.

## 8 Limitations and Broader Impact

In this work, we were primarily focused to establish the consistency of maximum likelihood estimates of the PPCA model in a quotient Euclidean space. However, we were neither able to provide a rate of convergence nor we could say something definitive about the asymptotic distribution of the MLE, and this opens up two avenues for future research. Nonetheless, it is worth noting that our research does not present any discernible negative societal implications.

## 9 Conclusion

In this paper, we proposed a novel topological framework to provide rigorous justification for the theoretical validity of the maximum likelihood estimation of the Probabilistic Principal Component Analysis (PPCA) model. Although our consistency results hold within a quotient space, implying that the true parameter is recoverable up to a closed subset of the parameter space, our results establish that this represents the optimal achievable outcome due to the challenges caused by identifiability inherent in the underlying model. Our result is stated in terms of the abstract framework of quotient topological spaces, but an immediate (and concrete) consequence of our work is (strong) consistent covariance estimation of the PPCA process through maximum likelihood estimates, which wasn't known before. Our rigorous framework opens doors for applications in many other statistical models where rotational ambiguity (or more generally ambiguity due to a closed space or symmetric parametrization) is present. One such example is the matrix factorization problem where one is interested in the estimation of two matrices $\mathbf{A}, \mathbf{B}$ such that the data matrix $\mathbf{X} = \mathbf{AB} +$ noise is one such problem. Our methodology is highly flexible in the sense that it does not depend on the statistical model much as long as there are some regularities in the geometry of the parameter space $\Theta$, which makes it even more usable, even for non-linear models such as nonlinear independent component analysis where the observed data is generated as $x = \xi(z|\theta) +$ noise, where $z$ is a latent random vector and $\theta$ is a parameter and the function $\xi$ is non linear. The methodology we developed in our work for the PPCA model could be readily applied in this case as our quotient space construction barely depends on the fact that $\xi$ is linear for PPCA. The primary focus for us was to build a strong theoretical foundation which was missing in the relevant literature. Furthermore, our study offers a concise and comprehensive treatment of the theory of quotient topological spaces, with an eye on statistical applications, building upon and expanding upon previous works with enhanced rigour and clarity.

## 10 Acknowledgements

The first author would like to thank his supervisors, Prof. Philippe Gagnon and Prof. Florian Maire, for many helpful conversations and continuous support by providing generous financial assistance and excellent working conditions during this project. The authors would like to express their gratitude to the anonymous reviewers and area chairs of NeurIPS 2023 for their careful review of the paper and their valuable comments and suggestions, which have greatly contributed to improving earlier versions of the manuscript. The authors would also like to thank Prof. Nik Weaver in the mathematics department at Washington University in St. Louis for an insightful private communication. The first author received financial support from his supervisors and fellowships provided by the Faculté des études supérieures et postdoctorales (FESP) and the Bourse d'exemption at the Université de Montréal during the course of this work. The first author is sincerely thankful to his family and remains deeply indebted to his special friend who goes by the nickname 1915131 for providing support of all kinds unconditionally throughout the duration of the work.

The second author would like to acknowledge the financial assistance and excellent working condition provided by Northwestern University.

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

## 11   Appendix

### 11.1   Counterexample: Quotient topology and the metric topology on the quotient parameter space are different.

It was pointed out in the introduction that special care is needed when talking about continuity and convergence in probability within the quotient parameter space $\Theta/C$ as there are two different topologies at play. In fact, a substantial challenge of our work was to meticulously address the technical issues that arise from the interplay between these two topologies. In Redner [1981], it was claimed that the MLE converges to the true parameter in the quotient topological space and this follows from the theory of quotient spaces which is not true in general, as it is untrue that the quotient metric behaves in a similar fashion to that of the original Euclidean metric and therefore it would take a great deal of effort to unambiguously define the right notion of convergence to be used for assumption 5 (and 3) in Redner [1981]. The interaction between the metric topology and the quotient topology depends highly on the equivalence relation $\sim$ on the parameter space $\Theta$. Here is a counter-example highlighting a situation where the quotient topology is nontrivial but the topology generated by the quotient metric $\bar{d}$ is degenerate:

Consider the space $X = \{(x, y) : x, y \geqslant 0\} - \{(0,0)\}$ with the usual Euclidean norm, i.e., the first quadrant with axes except the origin and define the equivalence relation $\sim$ on $X$ given by: $p_1 \sim p_2 \iff p_1 = \lambda p_2$ for some $\lambda > 0$, i.e., the points $p_1, p_2$ are equivalent if they are on the same ray. The projection map here is $\pi : X \to X/\sim$ is given by $\pi(p) = [p]$. Geometrically, it is helpful to visualize the space $X/\sim$ as a north-east part of the unit circle $\mathbf{S}^1$, which is a subspace of a full circle. Under this equivalence relation it is true that $d_1([x], [y]) = 0$ (from the definition of pseudo metric $d_1$ in section 4) for any two distinct $[x], [y] \in X/\sim$, which can be easily argued by taking $x_1 = x/n, y_1 = y/n$ with $n \in \mathbb{N}$, so that, $x_1 \sim x, y_1 \sim y$ and letting $n \to \infty$. Therefore the usual metric structure totally breaks down in the quotient space as any two different equivalence class has a distance zero between them while in this case the quotient topology is non-trivial and we can still talk about convergence with regards to the quotient topology. But to extend Wald's condition in the quotient, the space $X/\sim$ must be endowed with a non-trivial metric space structure, which is impossible in a situation like this.

We also note that the quotient space *had to* pass through a pseudometric to achieve the desired metric space structure in order to set up the ground for Wald's results to extend. In general, it is very hard to interpret something meaningful out of this intangible metric from a statistical point of view. However, when the quotient is done with respect to a closed set, one can provide a concrete tangible geometrical description of the quotient metric space through Lemma 4.3, which in our work has been done in the context of PPCA. The model PPCA is *not* important here, rather the *key* part is that: the quotient has to be with respect to a *nice enough equivalence* (for instance with respect to a closed set, which is the case in Redner's work, our work and numerous other statistical models where identifiablity arises). This *key technical but unavoidable* construction seems to have gone unnoticed in earlier works. Note that, it was imperative to come up with an explicit and tangible description of the metric in the quotient space as that plays a crucial role towards the proof of assumption 5 (Lemma 7.5 in our work) in Redner [1981], whose proof was assumed straightforward.

### 11.2   Counterexample: Theorem 4.2 is not sufficient to ensure the continuity of the lift with respect to the quotient metric $\bar{d}$.

We pointed out in section 6 that Lemma 6.1 is not sufficient to guarantee the continuity of the lift of the map $\psi$ used in the deduction of Theorem 5.9. Indeed it is possible that a continuous (and *not* Lipschitz) function $g : \Theta \to Y$ lifts to a function $\tilde{g} : \Theta/C \to Y$ which is no longer continuous with respect to the metric topology generated by $\bar{d}$ on the space $\Theta/C$. Here is a counterexample which highlights the situation:

Let $X$ be a subset of the unit square $[0,1]^2$ consisting of the origin and the points $a_n = (\frac{1}{n}, 0), b_n = (\frac{1}{n}, 1 - \frac{1}{n})$, and $c_n = (\frac{1}{n}, 1)$ for $n \geqslant 2$. We endow $X$ with the Euclidean metric inherited from $[0,1]^2$. We declare the following equivalence relation $\sim$ on $X$: we let $a_n \sim b_n$ for all $n$, and let $g : X \to \mathbb{R}$ be the function such that $g(0) = g(a_n) = g(b_n) = 0$, and $g(c_n) = 1$. In the quotient space $X/\sim$, the sequence $\{[c_n]\}$ converges to the origin, so the lift $\tilde{g}$ isn't continuous, as $\tilde{g}([c_n]) = 1$ for every $n$ and $g(0) = 0$. One may note that the function $g$ is not

Lipschitz in this example since the points $\{b_n\}$ and $\{c_n\}$ will become arbitrarily close as $n \to \infty$ while $|g(b_n) - g(c_n)| = 1$. Even though the domain of the function $g$, in this case, is discrete, one can extend $g$ continuously to $[0, 1]^2$, using the Tietz extension theorem (see Munkres) and provide a counterexample where the domain is an Euclidean space.

We note that $\psi$ (used in the deduction of Theorem 5.9) is not Lipschitz in the parameter space $\Theta$ due to the presence of a quadratic term in $\mathbf{W}$. One way to ensure that $\psi$ is Lipschitz is to assume the existence of a compact subset $\Theta_0$ which contains the point $\theta_0$ in its interior.

### 11.3  Proofs of auxiliary lemmas

*Proof of Lemma 7.1.* This is a direct consequence of the properties discussed in Section 4 and in particular, the fact that the topology generated from the metric $\bar{d}$ is weaker than the standard quotient topology on $\Theta/C$. Let $K$ be a closed and bounded set in $(\Theta/C, \bar{d})$ and $\{U_\alpha\}_{\alpha \in \Lambda}$ be an open cover for $K$. Since the set $\pi^{-1}(K)$ is closed (by the continuity of quotient map) and bounded in $\Theta$, it is compact. Therefore the open cover $\{\pi^{-1}(U_\alpha)\}_{\alpha \in \Lambda}$ can be reduced to a finite sub-cover of $\pi^{-1}(K)$ in $\Theta$. Using the surjectivity of the quotient map, we then see that there is a finite subset $F \subset \Lambda$ such that

$$K = \pi(\pi^{-1}(K)) \subseteq \pi\left(\bigcup_{\alpha \in F} \pi^{-1}(U_\alpha)\right) = \bigcup_{\alpha \in F} U_\alpha.$$

□

*Proof of Lemma 7.2.* To ensure the measurability conditions so that the above integrals are well defined we include all the $P_{\theta_0}$ null sets to the associated Borel sigma algebra as discussed in Bertsekas and Shreve [1978] (p.167, Corollary 7.42.1).

Given a parameter $\phi = (\mathbf{W}_\phi, \sigma_\phi^2) \in \Theta$, and $r > 0$, we note the function $\phi \mapsto -p\log(2\pi)/2 - \log \det(\mathbf{W}_\phi \mathbf{W}_\phi^T + \sigma_\phi^2 I)/2$ is continuous functions on $\Theta$. If $\phi \sim \phi'$, then the function evaluates to the same value for $\phi$ and $\phi'$. Therefore the function factors through the quotient topological space $\Theta/C$ by Theorem 4.2. In this case, the map $\phi \mapsto -p\log(2\pi)/2 - \log \det(\mathbf{W}_\phi \mathbf{W}_\phi^T + \sigma_\phi^2 I)/2$ can be realized as a continuous function on the quotient space $\Theta/C$. For $[\theta] \in \Theta/C$ fixed, we consider the ball $N_r([\theta])$, which is compact. Thus by the virtue of continuous functions, the map will have finite maximum in $N_r([\theta])$. We call these maximum $M_1(r)$. The log-likelihood $\log f(x; \phi)$ is given by

$$\log f(x; \phi) = -\frac{p\log(2\pi)}{2} - \frac{1}{2}\log \det(\mathbf{W}_\phi \mathbf{W}_\phi^T + \sigma_\phi^2 I_p) - \frac{1}{2}x^T(\mathbf{W}_\phi \mathbf{W}_\phi^T + \sigma_\phi^2 I_p)^{-1}x,$$

and thus we have that

$$\mathbb{E}_{\theta_0}[\log f^*(x, \theta, r)] = \int_{\mathbb{R}^p} \log \max\left\{1, \sup_{[\phi] \in N_r([\theta])} f(x; \phi)\right\} dP_{\theta_0}(x)$$

$$\leqslant \int_{\mathbb{R}^p} \sup_{[\phi] \in N_r([\theta])} \log f(x; \phi) dP_{\theta_0}(x)$$

$$= \int_{\mathbb{R}^p} M_1(r) dP_{\theta_0}(x) - \frac{1}{2}\mathbb{E}_\theta\left[\inf_{[\phi] \in N_r([\theta])} x^T(\mathbf{W}_\phi \mathbf{W}_\phi^T + \sigma_\phi^2 I)^{-1}x\right]$$

$$\leqslant M_1(r) \quad \left(\text{since } (\mathbf{W}_\phi \mathbf{W}_\phi^T + \sigma_\phi^2 I)^{-1} \text{ is positive definite}\right).$$

Bounding $h^*(x, s)$ is a similar task. First we note there exists $M^* > 0$ such that $f(x; \phi) < 1$ (using Wald's assumption 3 Wald [1949] or the next lemma) whenever $\bar{d}([\phi], 0) = \|[\phi]\| > M^*$. Pick $s$ sufficiently large such that $[\phi] \notin N_s([\theta_0])$ implies $\|[\phi]\| \geqslant M^*$ (can be ensured using triangle inequality $\|[\phi]\| \geqslant \|[\theta_0] - [\phi]\| - \|[\theta_0]\|$) and thus in this case $h^*(x, s) = 1$ and $\mathbb{E}[\log h^*(x, s)]$ evaluates to 0. □

*Proof of Lemma 7.3.* Since $\bar{d}([\theta_i], [\theta_0]) \to \infty$ in $\Theta/C$ implies $d(\theta_i, \theta_0) \to \infty$ in $\Theta$ (by definition of $\bar{d}$ in Section 4), we have $\|\theta_i\| \to \infty$ by triangle inequality. We aim to show $\log f(x; \theta_i) \to -\infty$. The

log-likelihood, therefore, is bounded by

$$
\begin{aligned}
\log f(x; \theta_i) &= -\frac{p \log(2\pi)}{2} - \frac{1}{2} \log \det(\mathbf{W}_{\theta_i} \mathbf{W}_{\theta_i}^T + \sigma_{\theta_i}^2 I_p) - \frac{1}{2} x^T (\mathbf{W}_{\theta_i} \mathbf{W}_{\theta_i}^T + \sigma_{\theta_i}^2 I_p)^{-1} x \\
&\leqslant -\frac{1}{2} \log \det(\mathbf{W}_{\theta_i} \mathbf{W}_{\theta_i}^T + \sigma_{\theta_i}^2 I_p) \\
&= -\frac{1}{2} \log \mathcal{P}_{-\mathbf{W}_{\theta_i} \mathbf{W}_{\theta_i}^T}(\sigma_{\theta_i}^2) \qquad \left( \mathcal{P}_{-\mathbf{W}_{\theta_i} \mathbf{W}_{\theta_i}^T} \text{ is the characteristic polynomial} \right) \\
&= -\frac{1}{2} \sum_{j=1}^{p} \log(\sigma_{\theta_i}^2 + \lambda_{ij}) \qquad \left( \{\lambda_{ij}\}_{j=1}^p \text{ eigenvalues of } \mathbf{W}_{\theta_i} \mathbf{W}_{\theta_i}^T \right) \\
&\leqslant -\frac{1}{2} \log \left( \sigma_{\theta_i}^2 + \|\mathbf{W}_{\theta_i}\| \right).
\end{aligned}
$$

We observe that if $\|\theta_i\| \to \infty$, either $\|W_{\theta_i}\|$ or $\|\sigma_{\theta_i}^2\|$ (or both) must become arbitrarily large, which in turn implies the required result. $\qquad \square$

*Proof of Lemma 7.4.*

$$
\begin{aligned}
\mathbb{E}_\phi[|\log f(x; \theta)|] &= \int_{\mathbb{R}^p} |\log f(x; \theta)| dP_\phi(x) \\
&= \int_{\mathbb{R}^p} \left( \frac{p \log(2\pi)}{2} + \frac{1}{2} \log \det(\mathbf{W}_\theta \mathbf{W}_\theta^T + \sigma_\theta^2 I) \right) dP_\phi(x) \\
&\quad + \frac{1}{2} \mathbb{E}_\phi[x^T (\mathbf{W}_\theta \mathbf{W}_\theta^T + \sigma_\theta^2 I)^{-1} x] \\
&\leqslant c_1(\theta) + \frac{1}{2} \mathbb{E}_\phi[\|x\|_2 \|(\mathbf{W}_\theta \mathbf{W}_\theta^T + \sigma_\theta^2 I)^{-1} x\|_2] \\
&\leqslant c_1(\theta) + \|(\mathbf{W}_\theta \mathbf{W}_\theta^T + \sigma_\theta^2 I)^{-1}\| \mathbb{E}_\phi[\|x\|_2^2].
\end{aligned}
$$

$\qquad \square$

*Proof of Lemma 7.5.* Suppose, $\theta \in C$. We observe $\bar{d}([\theta_i], [\theta]) \to 0$ in $\Theta/C$ implies $d(\theta_i, C) \to 0$ in $\Theta$. Since $C$ is a closed set and $\Theta$ is complete, therefore the sequence $\{\theta_i\}$ converges to some $\theta_0 \in C$. As $\theta \mapsto f(x; \theta)$ is continuous, we have $f(x; \theta_i) \to f(x; \theta_0) = f(x; \theta)$, in this case. If $\theta \notin C$, then there is a neighbourhood $U_\theta \subset \Theta$ of $\theta$ such that $U_\theta \cap C = \emptyset$ and consequently all but finitely many terms in the sequence $\{\theta_i\}$ belong to $U_\theta$. This implies, in this case, $\bar{d}([\theta_i], [\theta]) = d(\theta_i, \theta)$ for all but finitely many $i$ and the conclusion readily follows. $\qquad \square$