# OpenReview forum: "On the Consistency of Maximum Likelihood Estimation of Probabilistic Principal Component Analysis"
_NeurIPS.cc/2023/Conference — NeurIPS 2023 poster_

### Official Review · Reviewer_QWak · 2023-06-22

**Soundness:** 3 good
**Presentation:** 3 good
**Contribution:** 1 poor
**Rating:** 3
**Confidence:** 3

**Summary:**

This work studies the consistency of the maximum likelihood estimates of probabilistic PCA. These estimates are unique up to a rotation, so they use the quotient space and claim in their Lemma 6.1, ..., 6.5 stated in Redner[1981] are verified and apply the consistency results available in this book.

**Strengths:**

This paper is overall well written, with a thorough introduction to equivalence classes and quotient topological spaces.

**Weaknesses:**

Compared to Redner [1981] and Wald [1949], the novelty is not enough. I see the contribution as mainly an application of the result of Wald to the case of probabilistic PCA. The general idea is rather straightforward: using the quotient space instead of $R^d$ is the first idea that comes to mind. I agree that the precise formulation is not trivial but alone, and although the writing is excellent, it is not enough for a publication in my opinion.

**Questions:**

Could you clarify the contributions of the paper ? Is there something I missed ?

**Limitations:**

Not applicable.

---

> ### Author Rebuttal · Authors · 2023-08-08
>
> $\textbf{Respectful Disagreement}$
>
> Thank you for your careful review.
> We have clarified the details of our theoretical contributions and we respectfully disagree with the assessment that the contribution is mainly an application of the result of Wald and that there is inadequate reproducibility Not only are there significant technical novelties where our work differs from Redner's (1981), but our research definitely impacts future research}
>
> $\textit{Novel Contributions}$
>
> We agree that the quotient of $\mathbb{R}^d$ is the first that comes to mind but it takes a significant amount of technical novelty to rigorously build a framework that is amenable to an extension of Wald's work. The model PPCA has been around since 1999. In light of the widely used ML estimates it is highly unlikely that nobody tried to justify them through an argument based on Wald's criteria. The main difficulty was to interpret Wald's condition in a quotient space framework which partly has been introduced in Redner's work but that is neither precise nor complete.
>
> While Redner's work is the first one to $\textit{attempt}$ to address this issue with the help of quotient topology, the treatment developed by Redner is rather misleading as highlighted in our work (l67-78, l234-247). We humbly request the reviewer to glance through Redner's paper where he states Thereom 4. Unfortunately, it is assumptions 5 (and 3) that Redner takes for granted. For instance, it is simply untrue that the quotient metric behaves in a similar fashion to that of the original Euclidean metric and therefore it would take a great deal of effort to unambiguously define the right notion of convergence to be used for assumption 5 (and 3), which Redner seems to have missed. Redner mentions that the MLE converges to the true parameter in this topological space and this follows from the theory of quotient spaces which is not true. In general, metric topology and quotient topology are two very different things and their interaction depends highly on the equivalence relation $\sim$. Here is a counter-example: Consider the space $X=\{(x,y):x,y\geqslant 0\} - \{(0,0)\} $ with Euclidean norm, and define the equivalence relation $\sim$ on $X$ given by: $p_1\sim p_2\iff p_1=\lambda p_2$ for some $\lambda>0$, i.e., the points $p_1,p_2$ are equivalent if they are on the same ray. The projection map here is $\pi:X\to X/\sim$ is given by $\pi(p)=[p]$. Under this equivalence relation it is true that $d_1([x],[y])=0$ (from the definition of pseudo metric $d_1$ after l208 in paper) for any two distinct $[x],[y]\in X/\sim$, which can be easily argued by taking $x_1=x/n,y_1=y/n$ with $n\in\mathbb{N}$, so that, $x_1\sim x,y_1\sim y$ and letting $n\to \infty $.  Therefore the usual metric structure totally breaks down as any two different equivalence classes have a distance zero between them. But quotient topology is non-trivial and we can still talk about convergence there.
>
> To extend Wald's condition, the space $X$ must be endowed with a non-trivial metric structure which Redner has missed. For instance, it is impossible in the above example. The quotient space has to pass through a pseudometric to achieve the desired metric space structure in order to build set up the ground for Wald's results to extend. In general, it has been very hard to interpret something meaningful out of the intangible metric structure from a statistical point of view. However, when the quotient is done with respect to a closed set one can provide a concrete tangible geometrical description of the metric space through Lemma 4.3, which in our work has been done in the context of PPCA.
>
> The model PPCA is $\textit{not}$ important here, rather the $\textit{key}$ part is that the quotient has to be with respect to a $\textit {nice enough equivalence}$. This key technical but unavoidable construction seems to have gone unnoticed by Redner. Not only our contribution is building this connection that requires a significant amount of bridging between different equivalences that are governing the quotient space, but coming up with an explicit description of the metric in the quotient space is a significant contribution that was only ``existential" prior to us. This explicit description plays a crucial role in the proof of assumption 5 (our Lemma 6.5) which again was assumed trivial in Redner's work (see the proof of Theorem 5). Also, note that in that proof, it was taken for granted that $g^{*}(x,\gamma,r)$ is measurable, which is not true generally. It is only upper semianalytic (see proposition 7.47, p179 in the reference we gave in proof of Lemma 6.2).
>
> $\textbf{Impact and reproducibility for future research:}$
>
> As an immediate consequence of our work is (strong) consistent covariance estimation of the PPCA process which wasn't known before. (please see the second last paragraph of reviewer ZSYo's response) Our rigorous framework opens doors for many other general statistical models where rotational ambiguity (or more generally ambiguity due to a closed space or symmetric parametrization) is present. For instance, Matrix factorization problem where one is interested in the estimation of two matrices $\textbf{A},\textbf{B}$ such that the data matrix $\textbf{X}=\textbf{A}\textbf{B}+\text{noise}$ is one such problem, thanks to the Reviewer 1.
>
> Our methodology is highly flexible in the sense that it does $\textit{not}$ depend on the statistical model much as long as there are some regularities in the geometry of the parameter space $\Theta$. (please see the last paragraph of reviewer ZSYo's response)
>
> Our contribution includes an explicit formulation for smooth enough equivalence relations that was very much needed to make this flexible topological toolkit widely accessible to future researchers. Finally, we welcome your further suggestions and comments to improve the final manuscript.
>
> $\textit{We hope we have addressed all comments satisfactorily and kindly request you to revise your score.}$

---

> > ### Comment · Reviewer_QWak · 2023-08-16
> >
> > Thanks for your answer.
> > I agree that Redner's work was incomplete and that the current work fills this hole (in my opinion Redner's work is the reason why no one tried to prove consistency of PPCA before).
> > I agree that this method could be used for different models.
> > Nevertheless, once the mistake in Redner's work has been noticed, the distance in Lemma 4.3 is the first you would think of. Verifying Wald's assumptions does not pose any technical challenge as long as I can see. I keep my score unchanged.

---

> > > ### Author Response · Authors · 2023-08-16
> > >
> > > Thank you for your response and for engaging with us for a productive discussion. We would like to bring the following points to your attention with regard to Redner's work and the verification of Wald's criteria.
> > >
> > > i) We assume you meant Lemma 4.3. We respectfully disagree that the distance in that result is the first one that comes to anyone's mind and that is because it is not in general a distance on quotient parameter space $X/C$ (it is a distance on some other quotient space $X_{\approx}$). In our case, it happens to be a distance on $X/C$. This connection and why they are the same is far from being trivial. Please refer to our response to the first question of Reviewer gPBN where we exclusively discuss this point and consider looking at the reference provided there.
> > >
> > > ii) Redner (1981), `Note on the consistency of the maximum likelihood estimate for nonidentifiable distributions', Annals of Statistics is a fairly cited paper in the relevant literature. But if you look deeper into those citations, you will find it is mostly an acknowledgement saying that the situation has been handled within a quotient space (which is, unfortunately, a misconception). The methodologies were never used afterwards as the foundation was shaky. Therefore, it is important to have a correct foundation for it to be useful for future practitioners and clear up certain misconceptions. In our work, we tried our best to maintain neutral language while pointing out these issues. Contrary to Redner's work, our framework is not limited to an abstract piece of mathematics as it allows us to prove consistent covariance estimation as we pointed out in the response of the previous reviewer. Therefore, it is useful for practitioners and more experimentally inclined researchers for implementation purposes.
> > >
> > > iii) With regard to your last point, it is true that verifying Wald's conditions is far less challenging than actually building the theory in the correct way. With that said it has its challenges. As we said, in general, sup (or inf) over an uncountable family of functions gives rise to measurability issues which are hardly addressed (or even mentioned) in the previous works. Also in the third line of our proof of Lemma 6.3 (in the upper bound of log-likelihood), the short argument relied on the homoskedastic error term assumption for the PPCA model. Had this been a Factor model with heteroskedastic error term the proof would not have been the same. Therefore we tried to exploit the model assumptions on PPCA whenever we could and it is not merely a replication of something more general.

---

### Official Review · Reviewer_MhxU · 2023-07-04

**Soundness:** 3 good
**Presentation:** 3 good
**Contribution:** 3 good
**Rating:** 6
**Confidence:** 2

**Summary:**

In this work, the authors propose a novel topological framework and show that the maximum likelihood (ML) solution of probabilistic principal component analysis (PPCA) is consistent in an appropriate quotient Euclidean space. The consistency results encompass more estimators beyond the ML solution. In addition, the ML solution has been shown to achieve strong consistency when the parameter space is compact.

**Strengths:**

This work seems to be the first work to establish the (strong) consistency result of the ML solution of PPCA.

**Weaknesses:**

It would be ideal to remove the assumption that the latent dimension $q$ is known.

**Questions:**

I am not familiar with the techniques used in this work. Does the consistency in the appropriate quotient Euclidean space means that the ML estimate $(\hat{\mathbf{W}},\hat{\sigma}^2)$ converges to the true parameters $(\mathbf{W}_0, \sigma^2_0)$ in probability up to rotation (i.e., $\hat{\mathbf{W}}$ converges to $\mathbf{W}_0 \mathbf{R}$ with $\mathbf{R}$ being orthogonal)?

---

> ### Author Rebuttal · Authors · 2023-08-08
>
> $\textbf{We have addressed all the questions raised by the reviewer}.$
> Thank you for your careful review and assessment.
>
> $\textit{You are correct}$. If we assume $\sigma^2=\sigma_0^2$ is known then our result does imply that $\widehat{W}$ converges to $W_0R$ for some orthogonal matrix $R$, which is to say $\mathbb{P}([\widehat{W}]\to [{W}_0])=1$. However, it is not just an abstract and technical result. In this case, using standard results from probability theory it is possible to argue that: $\mathbb{P}(\widehat{W}\widehat{W}^T\to\widehat{W}_0\widehat{W}_0^T)=1$, as the function $W\to WW^T$ is continuous. This is a concrete result (and can be stated without referring to the quotient space) which says that the true covariance matrix $W_0W_0^T+\sigma_0^2I$ can be consistently estimated through the maximum likelihood estimates $\widehat{W}\widehat{W}^T+\sigma_0^2I$.
>
> On a slightly more technical side, note that our quotient space construction was necessary as an intermediate step in the previous argument, in fact, we crucially used the continuous function $W\to WW^T$ factors through the quotient space using Theorem 4.2 in our paper, we can give more details if you want but skip for now to keep the discussion simpler. Lastly, the known $\sigma^2$ assumption was only for illustration purposes and can of course be removed, and in that case, the parameter $(\widehat{W},\widehat{\sigma}^2)$ recovers $(W_0,\sigma_0^2)$ up to a closed set $C$ defined in our paper just above the line 229, and here $\widehat{W}\widehat{W}^T+\widehat{\sigma^2}I$ consistently estimates the covariance of the PPCA process, a theoretical guarantee with respect to the maximum likelihood estimate that was not known before.
>
> Finally, $\textit{regarding the weakness}$ you pointed out, thank you for raising this concern. We can remove the assumption that: $q$ is known. Our proof techniques work as long as $1\leqslant q\leqslant p$ is a fixed integer, i.e. they do not grow with the sample size $n$.
> Usually, it is a different line of interest to want to estimate $q$ (related to model selection problems), also $q$ can be estimated consistently through MLE; We discussed all these relevant works in lines 134-148 in our work, which motivated us not to focus too much on $q$. The authors warmly welcome further questions, suggestions to improve the current manuscript, or if there is anything that the reviewer would want to see in the final version. Due to the page constraint, we had to make a choice on the content to present and our main focus was to judiciously build a rigorous framework to make it available for the community which was only partly addressed before by the previous work, Redner (1981).
>
> $\textit{We hope we have addressed all your comments satisfactorily and kindly request you to revise your score.}$

---

> > ### Author Response · Authors · 2023-08-16
> > **Correcting a minor typo**
> >
> > Hello everyone,
> >    It has come to our notice that there is a minor (and obvious) typo in the response we wrote for reviewer MhxU. In the second line of the rebuttal it should have been $\mathbb{P}(\widehat{W}\widehat{W}^T\to W_0W_0^T)=1$ and not $\mathbb{P}(\widehat{W}\widehat{W}^T\to \widehat{W}_0\widehat{W}_0^T)=1$.

---

### Official Review · Reviewer_gPBN · 2023-07-06

**Soundness:** 3 good
**Presentation:** 4 excellent
**Contribution:** 3 good
**Rating:** 7
**Confidence:** 5

**Summary:**

The paper discusses consistency of the maximum likelihood (ML) estimation in probabilistic principal component analysis (PPCA). Despite its wide applicability, proving ML estimation consistency in PPCA has been a challenging task because of the non-identifiability of the problem. The author(s) extend the quotient space topology idea presented in Redner (1981) to a more general setting where ML estimator is one such estimator. Also, strong consistency of the ML estimator is proved under a compactness assumption.

**Strengths:**

1. Detailed description of the quotient space topology and the associated metric. The constructive approach was clear and concise.
2. With standard assumptions on the quotient space, the author(s) derived both weak and strong consistency of the ML estimator in the PPCA model.
3. Verification of the Wald's conditions [Wald, 1949]  in the quotient space topology with sufficient technical details.

**Weaknesses:**

1. Some remarks on the Wald's conditions on the quotient space to clarify how those conditions could be interpreted in the quotient space would be useful.
2. One or two examples to clarify the result in Theorem 4.2 would be helpful.

**Questions:**

1. What would be $X_{\equiv}$ of Lemma 4.3 in the case of PPCA and in what sense $X_{\equiv}$ and the quotient $X\C$ are "identical" in that lemma? The two metric defined on those two topological spaces are not same- right?

2. I was wondering whether it's not possible to have the consistency result for the MLE in PPCA as a corollary of the results proved in the following

Van der Vaart, A. W., & Wellner, J. A. (1992). Existence and consistency of maximum likelihood in upgraded mixture models. Journal of multivariate analysis, 43(1), 133-146.

I understand the above may be for mixture models but can't we have a corollary for one mixing component which would be the marginal distribution of the data given W and $\sigma$ in your case.

**Limitations:**

Yes, the author(s) have adequately addressed the limitations.

---

> ### Author Rebuttal · Authors · 2023-08-08
>
> $\textbf{We addressed all the questions raised by the reviewer.}$
>
> Thank you for your careful review and meticulous reading and assessment. We are very happy that you asked two excellent questions which we would like to address below:
>
> Let us clarify the picture in depth before answering the question, we recall a few things.
>
> i) We have a topological (and metric) space $X$ which we want to quotient wrt. an equivalence relation $\sim$.
>
> ii) Problem is there is no nice metric structure (in general) on $X/\sim$, which seems to have gone unnoticed in the previous work by Redner (1981). There is however, a well-known pseudometric (defined after line 208) and the topology this pseudometric generates can be very different from the quotient topology (please see the example provided in reviewer QWak's response). So we have to be very careful when saying things like convergence, and continuity as those concepts crucially rely on which topology we are using.
>
> iii) To extend Wald's condition we need a metric space as written towards the end of Wald's original paper. Therefore $X/\sim$ with a pseudometric is not enough. However this pseudometric can be turned into a metric through another equivalence relation $\approx$. (which basically treats two points `same' if their distance under this pseudometric is 0 and thus turns it into a metric).
>
> iv) Now there are two equivalene relations on $X$, namely $\sim$ and $\approx$ ($\approx$ and $\sim$ are interrelated as we mentioned in line 216). Therefore there are two possible quotient spaces $X/\sim$ endowed with a pseudometric and $X/\approx$ endowed with a metric. Of course, those spaces are related as $\sim$ and $\approx$ are related. But we want to work with $X/\approx$ since that has a metric space structure and we have a hope to check Wald's condition there.
>
> v) Notice $X/\sim$ (which is the same as $X/C$) is our parameter space where we should be trying to extend Wald's condition but in the previous point, we said we want to work in $X/\approx$ because $X/\sim$ might fail to have the desired metric space structure.
>
> vi) It is a non-trivial fact (please see p18, Lipschitz algebras, Nik Weaver) that for some `nice' (in our case which is a quotient by a closed set $C$) equivalence relations $\sim$, it is true that $\sim=\approx$ and the equality here means, they partition $X$ into same equivalence classes. The great news now is that: $X/\approx=X/\sim$, implies that the pseudometric on $X/\sim$ is actually a metric that we crucially exploit.
>
> $\textit{Now to answer your question}$,
>
> 1) The space $X_{\approx}$ is abstract and does not have a very clear picture in general but in our paper, it is essentially $X/C$ which has a clear picture. You are absolutely right that in general the metric structures are not the same but in our case it is, which is one of the key things for Wald's idea to work. Lastly, we mention that for most statistical applications (like Redner(1981)) we quotient by a closed space, and then we should be in a good situation. However, the reference we provided in point(vi) above outlines a fairly general framework beyond just quotient wrt a closed set, under which $\sim=\approx$ holds, and also gives solid counterexamples when any of those conditions are violated which again explains why care is needed to build a correct foundation.
>
> 2) Authors are aware of the work: Van der Vaart, A. W., \& Wellner, J. A. (1992). Our work won't be a corollary of their proven theorems as they assume model identifiability (see their assumption (3.3) in section 3, consistency).
>
> Finally, we cordially thank you again for reading our work in-depth and we will include an example or two along with a brief discussion on how Wald's condition could be interpreted in the Quotient space in the final version. Due to the space constraint of this conference, we had to make a choice to present the topics. With regards to the previous works along this line the authors strongly felt it is highly important to keep things very transparent and build a solid foundation. This is the reason why we kept every detailed argument as a part of the presentation. We had to sacrifice a bit on possible implications of our work, one of which is notably a (strong) consistent covariance estimation of the PPCA model (please see the second last paragraph of reviewer ZSYo's response). However, the argument can get slightly technical and we can elaborate more if you are interested but overall it is an immediate implication of the convergence in the quotient space. Lastly, we very much welcome your further suggestion to improve our final version.
>
> $\textit{We hope we have addressed all your questions satisfactorily and kindly request you to revise your score.}$

---

> > ### Comment · Reviewer_gPBN · 2023-08-15
> > **Response to author rebuttal**
> >
> > I thank the author(s) for their detailed explanations in the rebuttal. I am happy with the answers they provided to my queries. I have increased my rating to "accept".

---

> > > ### Author Response · Authors · 2023-08-17
> > >
> > > Thank you for your response and for engaging with us for a productive post-rebuttal discussion. We appreciate your cooperation.

---

### Official Review · Reviewer_ZSYo · 2023-07-07

**Soundness:** 3 good
**Presentation:** 4 excellent
**Contribution:** 2 fair
**Rating:** 5
**Confidence:** 4

**Summary:**

paper addresses "Probabilistic PCA" which stands for a setup where the vector of p observations x can be written as x = Wz + eps where eps stands for an additive (Gaussian, centered, iid) noise, the matrix W \in R^{p x q} is an unknown and z in R^q is the other unknown . Only the value of q is known in advance and the goal is to recover W and z and remove the noise.
The main challenge is to address the fact that the model is invariant w.r.t. applying a rotation R to W and z: Wz = (W R) (R' z) . Note that  (W R) & (R' z) have the same euclidean norms as W & z. The contribution is to propose an analysis in the quotient space of the vector space by the rotations.

**Strengths:**

The paper contains a rigorous consistency analysis of the PPCA problem with respect to rotation invariance

**Weaknesses:**

The main focus of the paper on the rotation invariance does not seem to be the biggest concern with PPCA or other related (see Questions)  methods.

**Questions:**

1. There is a broader set of problems where rotation invariance causes trouble, and that set of problem directly related to the PPCA problem considered: all problems where the observations / measurements are of the form <u_i, u_j> or <u_i, v_j>. This comprises PPCA, but also matrix factorization and embedding problems. Can authors discuss how their method can expand to those problems?
2. Is there a relation between rotation invariance and computational complexity? Can a rotation be a distractor in the gradient steps convergence of a first order method?

**Limitations:**

The focus seems to be relatively narrow and implications of the proven results not sufficiently emphasized.

---

> ### Author Rebuttal · Authors · 2023-08-08
>
> $\textbf{We addressed all the questions and weaknesses raised by the reviewer}$.
>
> Thank you for your careful review and valuable assessment. First, we address your questions.
> 1) You are correct that `There is a broader set of problems where rotation invariance causes trouble' and we agree that PPCA is just a part of that broad set. Our methodology and framework are readily applicable even in those cases. To continue with your question, let us assume that (for matrix factorization) we are interested in the estimation of two matrices $\textbf{A},\textbf{B}$ such that the data matrix $\textbf{X}=\textbf{A}\textbf{B}+\text{noise}.$
> Assume that $X\in\mathbb{R}^{n\times p},A\in\mathbb{R}^{n\times m},B\in\mathbb{R}^{m\times p}$. Observe that if $(A, B)$ is a solution to $X=AB$ then so is $(AR, R^TB)$, where $R$ is any $m\times m$ orthogonal matrix. Let $A_0,B_0$ are the true values of $A$ and $B$, respectively. We can try to check Wald conditions on the space $\mathbb{R}^{n\times m}/\sim_{A_0}\times \mathbb{R}^{m\times p}/\sim_{B_0}$, where $\sim_{A_0},\sim_{B_0}$ are equivalence relations defined by $\sim_{A_0}:=\{ A' :A'A'^T=A_0A_0^T \}$  and $\sim_{B_0}=\{B': B'^TB'=B_0^TB_0\}$. Note that $\sim_{A_0},\sim_{B_0}$ are both closed subspaces of Euclidean space. Geometrically we are in a nice situation here. In fact, it will work out as long as the likelihood function for this matrix factorization model has some regularities. More generally, if the parameters $u_i,v_j$ are of your interest and the likelihood contains something like $\langle u_i,v_j\rangle$, you can always do the above construction.
> 2) Authors are unsure how this question is related to our work as we have a `closed form' expression of the Maximum likelihoods in our case stated on page 4 in the description of the PPCA model. We usually apply numerical schemes (like gradient descent) where closed forms are $\textit{not}$ available.
>
> Having said that, we respond to the question. If rotational invariance (or more generally model identifiability) is present then that will lead to too many (possibly uncountable) local minima which is bad news. Too many local minima might cause trouble especially if their cost is higher than the global minima of the objective function. But, it seems that even in this case, it is possible to treat those large number of local minimas equivalent with respect to their costs and do a similar kind of analysis but that would mean trying to understand the dynamics of gradient steps convergence of a first-order method on a quotient space (which is non-Euclidean).  It is possible though, to use our framework even in this case. But usually, to understand the dynamics, we need some sort of a smoothness (or a notion of differentiability) assumption beyond continuity which seems difficult on the quotient spaces. It is possible to do differential calculus on quotient spaces if they are manifolds and admit smooth structures. However, we can prove that those quotient spaces are not manifolds. So, standard methodologies from differential geometry do not apply. This seems like an excellent (but considerably harder due to reasons we mentioned) follow-up line of research in the future but for now is not very related to the work we have done.
>
> Lastly, we respectfully disagree with the $\textit{weakness}$ pointed out by the reviewer for the following reasons:
>
> 1) Authors understand that their result is stated in terms of abstract topological spaces but it readily implies consistent covariance estimation is possible. Though the PPCA community does not care much about rotational invariance they do care about consistent covariance estimation which was not known before. For instance our result implies $\mathbb{P}(\widehat{W}\widehat{W}^T+\widehat{\sigma^2}I\to W_0W_0^T+\sigma_0^2I)=1$ This follows from a standard fact in probability theory that $\mathbb{P}(X_n\to c)=1\implies \mathbb{P}(f(X_n)\to f(c))=1$ if $f$ is nice (continuous). In our case, the map $[(W,\sigma^2)]\to WW^T+\sigma^2I_p$ can be seen as a continuous map from the quotient space $X/C$ to the space $R^{p\times p}$ by Theorem 4.2.
> 2) As you already observed in the answer to the first question our methodology is highly flexible in the sense that it does $\textit{not}$ depend on the statistical model much as long as there are some regularities in likelihood function and in the geometry of the parameter space $\Theta$. This makes it even more usable, even for non-linear models like nonlinear independent component analysis where the observed data is generated as $x=f(z|\theta)+\text{noise}$, where $z$ is a latent random vector and $\theta$ is a parameter and the function $f$ is $\textit{non-linear}.$ The methodology we developed in our work for the PPCA model could be readily applied in this case as our quotient space construction barely depends on the fact that $f$ is linear for PPCA. It would be an interesting line of work to investigate the connection between the methodologies discussed in our work to that of Zheng et al `On the Identifiability of Nonlinear ICA: Sparsity and Beyond', NeurIPS 2022, where authors recover the latent sources up to an equivalence but using other techniques.
> There are numerous possible ways of research directions that can come out of this work. The primary focus for us was to build a strong theoretical foundation (that was missing in the literature, Redner(1981)), This allows seamless and flexible applications. Finally, we hope to make abstract topological ideas more mainstream and practical. For your assessment of us not sufficiently emphasising the implication of our work, please refer to the last paragraph of reviewer gPBN's response where we discuss this point. We warmly welcome your suggestions and comments to improve our final version.
>
> $\textit{We hope we have addressed all comments satisfactorily and kindly request you to revise your score.}$

---

### Decision · Program_Chairs · 2023-09-21

**Decision:**

Accept (poster)

**Comment:**

While it was pointed out that the contribution in techniques for obtaining the results may be limited, the reviewers agreed that the paper addresses something that has been overlooked in the literature, and the results constitute a useful contribution to the PPCA problem. Please incorporate the reviewers' comments when preparing the camera-ready version.